# The Impact of Core Tissues on Successful Next-Generation Sequencing Analysis of Specimens Obtained through Endobronchial Ultrasound-Guided Transbronchial Needle Aspiration

**DOI:** 10.3390/cancers13235879

**Published:** 2021-11-23

**Authors:** Keigo Uchimura, Komei Yanase, Tatsuya Imabayashi, Yuki Takeyasu, Hideaki Furuse, Midori Tanaka, Yuji Matsumoto, Shinji Sasada, Takaaki Tsuchida

**Affiliations:** 1Department of Endoscopy, Respiratory Endoscopy Division, National Cancer Center Hospital, 5-1-1, Tsukiji, Chuo-ku, Tokyo 104-0045, Japan; kyanase@ncc.go.jp (K.Y.); timabaya@ncc.go.jp (T.I.); ytakeyas@ncc.go.jp (Y.T.); hfuruse@ncc.go.jp (H.F.); midotana@ncc.go.jp (M.T.); yumatsum@ncc.go.jp (Y.M.); sasastaf@hotmail.co.jp (S.S.); ttsuchid@ncc.go.jp (T.T.); 2Department of Cardiology and Respiratory Medicine, Graduate School of Medicine, Gifu University, 1-1, Yanagida, Gifu 501-1194, Japan; 3Department of Thoracic Oncology, National Cancer Center Hospital, 5-1-1, Tsukiji, Chuo-ku, Tokyo 104-0045, Japan

**Keywords:** bronchoscopy, lung cancer, next-generation sequencing, endobronchial ultrasound-guided transbronchial needle aspiration (EBUS-TBNA)

## Abstract

**Simple Summary:**

Next-generation sequencing (NGS) with specimens obtained through endobronchial ultrasound-guided transbronchial needle aspiration (EBUS-TBNA) has been used to identify cancer-related genes among patients with lung cancer. However, the reported success rates vary, and the clinical factors associated with successful NGS remain unclear. We retrospectively reviewed consecutive patients with non-small-cell lung cancer who underwent EBUS-TBNA for NGS (Oncomine^TM^ Dx Target Test). The numbers of punctures and core tissues, as well as computed tomography (CT) and EBUS findings, were evaluated. The success rate of NGS was 83.3% (130/156). The cut-off value for the number of core tissues was 4, and the sensitivity and specificity for successful NGS were 73.8% and 61.5%, respectively. CT and EBUS findings were not associated with successful NGS. In logistic regression analysis, the number of core tissues (≥4) was the sole predictor of successful NGS. Bronchoscopists should obtain sufficient core tissues for successful NGS using EBUS-TBNA specimens.

**Abstract:**

The success rate of next-generation sequencing (NGS) with specimens obtained through endobronchial ultrasound-guided transbronchial needle aspiration (EBUS-TBNA) among patients with lung cancer as well as the related clinical factors remain unclear. We aimed to determine the optimal number of punctures and core tissues during EBUS-TBNA for NGS in patients with non-small-cell lung cancer (NSCLC) as well as the association of chest computed tomography (CT) and EBUS findings with successful NGS. We retrospectively reviewed 156 consecutive patients with NSCLC who underwent EBUS-TBNA for NGS (Oncomine^TM^ Dx Target Test). Using the receiver operating characteristic curve, we calculated the optimal numbers of punctures and core tissues for NGS and evaluated CT and EBUS findings suggestive of necrosis and vascular pattern within the lesion. The success rate of NGS was 83.3%. The cut-off value for the number of core tissues was 4, and the sensitivity and specificity of successful NGS were 73.8% and 61.5%, respectively. Logistic regression analysis revealed that the number of core tissues (≥4) was the sole predictor of successful NGS. CT and EBUS findings were not associated with successful NGS. Bronchoscopists should obtain sufficient core tissues for successful NGS using EBUS-TBNA specimens.

## 1. Introduction

Various molecular-targeted drugs for treating patients with advanced-stage non-small-cell lung cancer (NSCLC) that are tailored for specific driver gene mutations, including epidermal growth factor receptor gene (*EGFR*), fusions of echinoderm microtubule-associated protein-like 4 and anaplastic lymphoma kinase (*EML4/ALK*), c-ros oncogene 1 (*ROS1*), and v-raf murine sarcoma viral oncogene homolog B1 (*BRAF*), improve prognosis better than conventional cytotoxic chemotherapy [1,2,3,4,5]. Molecular target tests for gene mutations using tumor tissues, including polymerase chain reaction amplification, immunohistochemistry, and fluorescence in situ hybridization, are crucial for selecting the optimal treatment strategy for patients with NSCLC [6]. Previously, companion diagnostic tests for each gene mutation were used; however, next-generation sequencing (NGS) has recently allowed simultaneous searching of various gene mutation types [7,8].

In 2017, the US Food and Drug Administration approved the Oncomine^TM^ Dx Target Test (ODxTT; Thermo Fisher Scientific, San Jose, CA, USA) as an NGS-based companion diagnosis for NSCLC. Using as little as 10 ng of DNA or RNA, it can simultaneously evaluate 46 cancer-related genes [5,9,10,11,12,13,14]; further, it facilitates companion diagnosis of the following driver oncogenes: *EGFR*, *EML4/ALK*, *ROS1*, and *BRAF*. The ODxTT multi-companion diagnostics (CDx) system is a common NGS test covered by insurance in Japan. Compared with previous companion diagnostic tests, NGS requires tumor tissues with sufficient quantity and quality, including tumor content ratio and the number of tumor cells in the specimens. Moreover, recent studies have reported on the success rate of NGS using bronchoscopic specimens [8,10,11,12,13,14,15,16,17,18,19].

Endobronchial ultrasound-guided transbronchial needle aspiration (EBUS-TBNA) is a minimally invasive and standardized technique for sampling from mediastinal and hilar lesions that can be bronchoscopically approached from airways [6]. Two previous meta-analyses reported that EBUS-TBNA has a cumulative sensitivity and specificity of 88–93% and 100%, respectively, for staging NSCLC [20,21]. NGS using EBUS-TBNA specimens, which are needle biopsies, is impeded by tumor cell degeneration, blood contamination, and tumor tissue necrosis in the specimens [10,11,22]. Furthermore, EBUS-TBNA specimens facilitate NGS with a success rate of 46.0–95.3% [14,16,18,19]. However, these reported success rates vary, and the clinical factors associated with successful NGS remain unclear.

For EBUS-TBNA procedures, the American College of Chest Physicians’ (ACCP) guidelines suggest a minimum of three separate needle passes per target lesion for lesion diagnosis in patients with suspected lung cancer, as well as additional sampling for molecular analysis [23]. However, the optimal number of punctures and thread-like specimens obtained through EBUS-TBNA (and thus called “core tissue” (Figure 1)) for NGS remains unclear. Numerous studies have compared EBUS-TBNA puncture needle sizes across diagnostic parameters [22,24]; however, few studies have examined the success rate of NGS with each puncture needle size [16].

Bronchoscopists can routinely evaluate mediastinal and hilar target lesions using B-mode sonographic findings and color/power Doppler mode ultrasound images during EBUS-TBNA. Each sonographic finding allows differentiation between benign and malignant lesions [25,26,27]. Specifically, B-mode findings allow the assessment of the echogenicity of the lesion and the presence of necrosis, while color/power Doppler mode ultrasound images facilitate the assessment of the presence of abundant blood flow in the target lesion. Additionally, the degree of necrosis of the target lesion can be assessed on chest contrast-enhanced computed tomography (CT) before EBUS-TBNA [28]. However, it remains unclear whether these imaging findings of necrosis and blood flow within the lesion are associated with the success rate of NGS using EBUS-TBNA specimens.

We aimed to determine the optimal number of punctures and core tissues during EBUS-TBNA for NGS in patients with NSCLC, as well as whether chest CT and EBUS findings before and during EBUS-TBNA could predict the success of NGS (ODxTT multi-CDx system).

## 2. Methods

### 2.1. Patients

This single-center retrospective study included patients with primary or recurrent NSCLC who underwent ODxTT multi-CDx system analysis using EBUS-TBNA specimens from July 2019 to October 2020 at the National Cancer Center Hospital, Tokyo, Japan. We obtained approval from the institutional review board of the National Cancer Center Hospital, Tokyo, Japan (No. 2018-090); moreover, all the participants provided written informed consent for EBUS-TBNA. EBUS-TBNA and ODxTT multi-CDx system were performed when clinically required rather than solely for this study.

### 2.2. EBUS-TBNA Procedure

All patients underwent flexible bronchoscopy under moderate-to-deep sedation with intravenous anesthesia. EBUS-TBNA was performed using a convex probe ultrasound bronchoscope (BF-UC260FW or BF-UC290F, Olympus, Tokyo, Japan) and a 22-gauge (G) needle (NA-201SX-4022, NA-401SX-4022, Olympus, Tokyo, Japan; Expect^TM^ Pulmonary, Boston Scientific Corporation, Natick, MA, USA; EchoTip Ultra^®^ or EchoTip ProCore^®^, Cook Medical, Bloomington, IN, USA) or a 25-G needle (NA-U401SX-4025N, Olympus, Tokyo, Japan). Although 22-G needles were mainly used for EBUS-TBNA, 25-G needles were selected for cases that required systemic staging or for those in which the operator believed puncturing the tissue would be too difficult using a 22-G needle. Furthermore, a dedicated ultrasound processor (EU-ME2 PREMIER, Olympus, Tokyo, Japan) was used to evaluate B-mode findings and vascular pattern images with the color/power Doppler mode ultrasound at its maximum diameter. All procedures were performed under the supervision of two bronchoscopists with more than 10 years of experience.

The bronchoscope was inserted orally, and systemic staging of N1–N3 stations was performed in cases where the treatment strategy depended on the EBUS-TBNA results. In cases that did not require systemic staging, EBUS-TBNA was performed mainly on the lesions suggestive of malignancy by CT or EBUS findings. After confirming the absence of large blood vessels using color/power Doppler ultrasound at the puncture site, targeted lesions were punctured using stylets. EBUS-TBNA was performed using a negative pressure syringe (−10 or −20 mL) with up to 30–50 strokes per puncture, if there was a lot of blood backflow, and lower negative pressure or using the slow-pull method for the subsequent passes. The specimen within the needle was pushed out with an air-filled syringe onto glass slides. The obtained core tissues on the slides were immediately fixed with 10% neutral-buffered formalin for 6–48 h and embedded in paraffin. The slide from which the core tissue was removed was pressed onto another glass slide, and the two slides were fixed with 95% alcohol and treated with Papanicolaou stain. When performing rapid on-site evaluation (ROSE), one of the slides was dry-fixed and stained with Diff-Quik stain. In all patients, the first pass at each lesion was performed using ROSE. The numbers of punctures and core tissues (Figure 1) were counted per punctured lesion.

### 2.3. The Oncomine^TM^ Dx Target Test Multi-CDx System

The ODxTT multi-CDx system can analyze mutations at hotspots in 46 cancer-related genes using DNA derived from tumor specimens, as well as fusions in 21 cancer-related genes using RNA derived from tumor specimens with amplicon sequencing (Table 1).

ODxTT multi-CDx system was performed on 10–15 slides of 5 µm-thick sliced tissues prepared using formalin-fixed paraffin-embedded (FFPE) specimens containing core tissues obtained through EBUS-TBNA. DNA and RNA extractions were performed using the Ion Torrent Dx FFPE Sample Preparation Kit, which was included as part of the ODxTT multi-CDx system for the Applied Biosystems VRTi Dx (Thermo Fisher Scientific). The deparaffinized samples from FFPE specimens were lysed using protease and filter cartridges were used to separate DNA and RNA. DNA and RNA inputs were 10 ng each and were measured using the NanoDrop spectrophotometer (Thermo Fisher Scientific) and the Qubit^®^ 2.0 fluorometer (Thermo Fisher Scientific). Library preparation was performed using the ODxTT multi-CDx system, controls, and Ion PGM Dx Library Kit. Template preparation was performed using Ion One Touch Duo Dx and Ion One Touch Dx Template Supplies, and Ion PGM Dx Sequencing Kit was used for sequencing platform. Mutations among cancer-related genes were analyzed by the Torrent Suite Dx software (Thermo Fisher Scientific).

The ODxTT multi-CDx system is only approved for companion diagnostic testing for four cancer-related genes (*EGFR*, *EML4/ALK*, *ROS1*, and *BRAF*) in Japan. Therefore, in this study, “success” was defined as cases where all four cancer-related genes (*EGFR*, *EML4/ALK*, *ROS1*, and *BRAF*) could be analyzed through NGS. “Failure” was defined as cases diagnosed as NSCLC (by pathologists) that did not undergo NGS due to insufficient tumor tissue as judged by the pathologists (tumor content ratio <30 % and the number of tumor cells per slide <5000 cells) [10,16,29] and those in which the presence or absence of the four cancer-related genes (*EGFR*, *EML4/ALK*, *ROS1*, and *BRAF*) could not be analyzed.

### 2.4. EBUS Findings and Pathological Diagnosis

The B-mode findings and vascular pattern images were reviewed by two bronchoscopists with more than 10 years of experience who were blinded to the results of the pathological diagnoses.

Based on a previously reported B-mode image classification [25,27], images were evaluated according to the following parameters: size (short axis diameter >1 cm or ≤1 cm), shape (round or oval), echogenicity (heterogeneous or homogeneous), margins (distinct or indistinct), central hilar structure (presence or absence), and coagulation necrosis sign (presence or absence). The round shape was defined as a ratio of the short to long axis diameter < 1.5. The distinct margin was defined by clear visibility of the margin between the lesion and surrounding tissues (i.e., >5.0%). The central hilar structure was defined as a flat, linear, and hyperechoic region in the center of the lesion. The coagulation necrosis sign was defined as a hypoechoic region within the lesion that lacked blood flow. Based on a previous classification, we divided vascular patterns using color/power Doppler mode ultrasound into the following four grades [26]: Grade 0: no blood flow or small flow amounts; Grade 1: a few main vessels running toward the center of the lesion from the hilum; Grade 2: a few punctiform- or rod-shaped flow signals and a few small vessels found as a long, curved strip; and Grade 3: rich flow, more than four vessels with different diameters and twist or helical-flow signal. Grades 2–3 were distinguished from grades 0–1 by their abundant blood flow in the lesion. Each sonographic feature was analyzed according to the success of the NGS.

An independent cytologist and pathologist evaluated histological and cytological specimens. The final diagnosis of each lesion was based on the pathological evidence obtained from EBUS-TBNA or surgery.

### 2.5. Statistical Analysis

Data are presented as frequency, median (range), and percentage. Based on a two-sided hypothesis, between-group comparisons were performed using the Fisher’s exact test or Mann–Whitney *U* test. Receiver operating characteristic curve (ROC) analysis and the Youden index (sensitivity + specificity − 1) were used to calculate the optimal cut-off values for the numbers of punctures and core tissues. The trend was examined using the Cochran–Armitage test with one-sided *p*-values. Furthermore, we performed a multivariable logistic regression analysis of the clinical factors that were shown to influence the success of NGS in the univariable analysis. Statistical analyses were performed using EZR [30] (Saitama Medical Center, Jichi Medical University, Saitama, Japan), which is a graphical user interface for R (The R Foundation for Statistical Computing, Vienna, Austria, v2.13.0), and a modified version of R commander (v1.8-4, McMaster University, Hamilton, ON, Canada).

## 3. Results

### 3.1. Patients and Target Lesions

During the study period, 483 patients underwent EBUS-TBNA, with NSCLC being identified in 290 patients. ODxTT multi-CDx system analysis with EBUS-TBNA specimens was not ordered when a complete resection specimen had been ultimately obtained or when molecular analysis was not required. Finally, 156 patients underwent ODxTT multi-CDx system analysis with EBUS-TBNA specimens. Successful and failed NGS analyses were performed in 130 (83.3%) and 26 (16.7%) patients, respectively (Figure 2). NGS was not ordered for 12 patients due to insufficient pathological quantity of the specimens. Among the 144 patients whose specimens were considered to be appropriate for NGS by pathologists, “success” was achieved in 130 (90.3%) patients.

Table 2 summarizes the baseline characteristics of the patients and target lesions. There were no significant between-group differences in the patient characteristics.

### 3.2. Numbers of Punctures and Core Tissues for Successful NGS

Figure 3 shows the number of punctures and core tissues in both groups. The success cases showed a higher number of punctures (4 vs. 5, *p* = 0.021) than the failure cases (Figure 3A). Using the ROC curve, with a cut-off value of four times, the sensitivity and specificity for predicting successful NGS were 38.5% and 86.9%, respectively. The area under the curve (AUC) was 0.637 (95% confidence interval (CI), 0.51–0.765) (Figure 3B). Success cases showed a higher number of core tissues (3 vs. 4, *p* < 0.001) than the failure cases (Figure 3C). Using the ROC curve, with a cut-off value of four core tissues, the sensitivity and specificity for predicting successful NGS were 61.5% and 73.8%, respectively. The AUC was 0.718 (95% CI, 0.613–0.823) (Figure 3D).

Figure 4 demonstrates the success rate of NGS using EBUS-TBNA specimens based on the number of punctures and core tissues. Higher numbers of punctures and core tissues were significantly associated with successful NGS (*p* = 0.028 and *p* < 0.001). Additionally, all eight patients for whom six core tissues were obtained through EBUS-TBNA had successful NGS.

### 3.3. Clinical Factors Affecting Successful NGS

Table 3 summarizes the success rate of NGS using specimens obtained through EBUS-TBNA according to clinical factors (EBUS-TBNA procedures, EBUS findings, and CT findings). Univariable analysis revealed that the number of punctures and core tissues (≥4) (*p* < 0.001 and *p* = 0.004) could predict successful NGS. Needle size, performing systemic staging, necrotic CT findings, B-mode findings, and vascular pattern images were not associated with successful NGS. Logistic regression analysis revealed that the number of core tissues (≥4) was the only independent predictor of successful NGS with EBUS-TBNA specimens (odds ratio, 3.39; 95% CI, 1.09–10.6; *p* = 0.035).

## 4. Discussion

This is the first study to show that the number of core tissues has the greatest contribution to the success rate of NGS (ODxTT multi-CDx system) using EBUS-TBNA specimens. The cut-off value for the number of core tissues required for successful NGS was 4. Moreover, the number of punctures and core tissues was positively correlated with the success rate of NGS. However, successful NGS could not be predicted by differences in the needle size, EBUS, and CT findings suggestive of necrosis or blood flow within the lesions.

ACCP guidelines regarding EBUS-TBNA procedures without ROSE suggest a minimum of three separate needle punctures per target lesion for diagnosing lesions in patients with suspected lung cancer, as well as additional sampling for molecular analysis [23]. A retrospective study reported that four puncture times could yield sufficient specimens for limited molecular tests (*EGFR*, v-Ki-ras2 Kirsten rat sarcoma viral oncogene homolog (*KRAS*), and *EML4/ALK*) [31]. Although the panels used for NGS analysis differ across studies, several studies have reported high success rates (>80%) of NGS using EBUS-TBNA specimens, with the median number of puncture times ranging from 4 to 6 [16,18]. On the other hand, a prospective study reported a success rate of NGS of only 46%, with an average of two puncture times during EBUS-TBNA [17]. Therefore, performing ≥ four punctures is essential for successful NGS using EBUS-TBNA specimens.

In addition, the number of core tissues was a more predictive factor for successful NGS than the number of punctures. We previously reported that in 29.0–41.4% of single EBUS-TBNA specimens obtained using a 22-G needle, only blood, cartilage, or bronchial walls were obtained, which rendered them unsuitable for diagnosis [24]. Similarly, two studies reported that 50–60% of EBUS-TBNA specimens obtained using stainless steel EBUS-TBNA needles were unsuitable for diagnosis when evaluated per puncture [32,33]. Collection of the core tissue during EBUS-TBNA can be confirmed macroscopically, which could indicate whether the sampled tissue is acceptable for NGS and pathological examination. However, the optimal number of core tissues for successful NGS using EBUS-TBNA specimens remained unclear. Our findings suggest that to allow successful NGS, bronchoscopists should ensure a sufficient number of tissues (≥4) is available during EBUS-TBNA.

We observed no differences in the success rate of NGS between EBUS-TBNA specimens obtained using 22- and 25-G needles, which is consistent with previous reports [16]. 22-G needles have a larger inner diameter than 25-G needles; therefore, they could be more suitable for tissue sampling. Several retrospective studies have demonstrated no differences in specimen adequacy or diagnostic accuracy between 22- and 25-G needles; moreover, given the structural advantage of 25-G needles, they have higher flexibility and penetrability than 22-G needles [34,35,36]. Most bronchoscopists consider using a 25-G needle for lesions when it is technically difficult to perform EBUS-TBNA (e.g., 4L lymph node). Our findings suggest that there is no need to limit the choice of needle size in patients requiring NGS of EBUS-TBNA specimens.

We found that B-mode findings, vascular pattern images on EBUS, and necrotic findings on CT could not predict successful NGS using EBUS-TBNA specimens. However, these findings do not imply that EBUS-TBNA procedures suitable for NGS may be performed without considering findings suggestive of necrosis or/and abundant blood flow in the target lesions. Although EBUS-TBNA is generally a safe procedure with a low complication rate, there have been reports of EBUS-TBNA-related complications, including bleeding, infections, and pneumothorax [20,37,38,39]. Recently, a large case-control study reported that necrotic findings within target lesions on enhanced chest CT were independently associated with infectious complications [40]. Therefore, punctures during EBUS-TBNA should be avoided in the target lesion’s area in the case of obvious necrotic findings on chest CT and B-mode sonography to allow adequate tissue sampling and avoid infectious complications [25]. Additionally, color/power Doppler ultrasound is used to avoid puncturing the target lesion’s area in the case of abnormally abundant or absent blood flow [26]. Our results demonstrated that NGS could be performed even for EBUS-TBNA specimens obtained from target lesions with necrosis or/and abundant blood flow if such basic puncture site selection is performed.

NGS panels vary according to institution and country. We used ODxTT, which is a relatively small-panel NGS testing method that can analyze 46 cancer-related genes [5,9,10,11,12,13]. A previous retrospective study reported no difference in the success rates of NGS panel testing for 50 and 1213 cancer-related genes (98% and 91%, respectively) using cytology specimens obtained using EBUS-TBNA [16]. On the other hand, large-panel NGS testing (>200 genes) requires larger DNA amounts (approximately 50 ng) [41], unlike ODxTT, which can analyze with 10 ng of DNA [10,29]. Notably, the optimal numbers of punctures and core tissues will vary according to the type of NGS panel testing.

This study has some limitations. First, this was a single-center retrospective study with a relatively small sample size. Second, all procedures were not performed by the same bronchoscopist; therefore, EBUS-TBNA specimens may have been affected by the skill of the bronchoscopists. Third, previous studies [14,15,16,17,18] on the success rates of NGS using EBUS-TBNA specimens have mostly used cytology smears or FFPE cell blocks rather than FFPE-containing core tissues. Sample processing may have influenced our findings. Fourth, this study was performed in clinical practice; moreover, we did not assess the concentration of DNA and RNA. Fifth, the results of the ROC curve analysis of the number of punctures and core tissues were insufficient. However, no clinical factors contributing to the success of NGS other than the number of core tissues were evident in this study. Therefore, the number of core tissues obtained through EBUS-TBNA should be focused on for ensuring successful NGS. Finally, we did not determine the success rate in patients for whom NGS was not ordered. Our findings should be confirmed by future prospective, multi-center trials.

## 5. Conclusions

In conclusion, the number of core tissues was the most important contributing factor to the success of NGS (ODxTT multi-CDx system) using EBUS-TBNA specimens. The success rate was positively correlated with the number of punctures and core tissues. Bronchoscopists should perform a sufficient number of punctures and obtain ≥4 core tissues for successful NGS using EBUS-TBNA specimens.

## Figures and Tables

**Figure 1 cancers-13-05879-f001:**
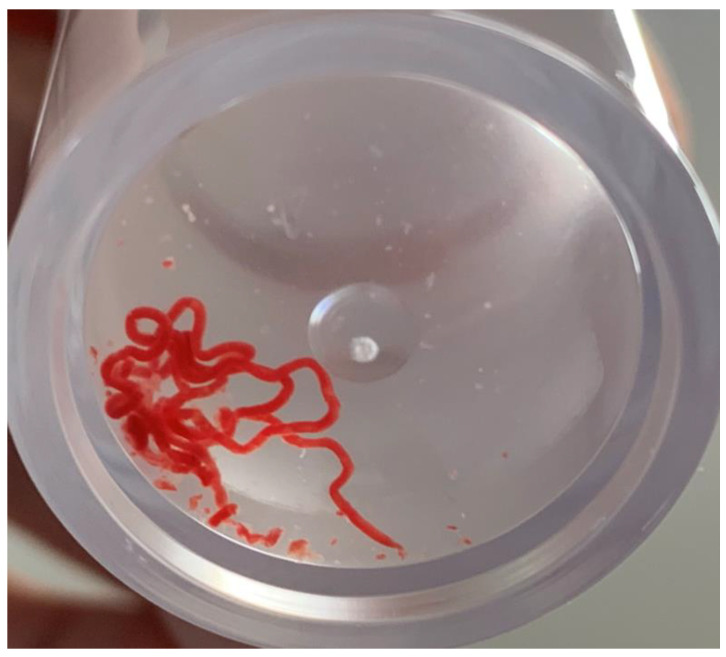
A thread-like specimen called “core tissue” obtained by a single endobronchial ultrasound-guided transbronchial needle aspiration in a formalin bottle.

**Figure 2 cancers-13-05879-f002:**
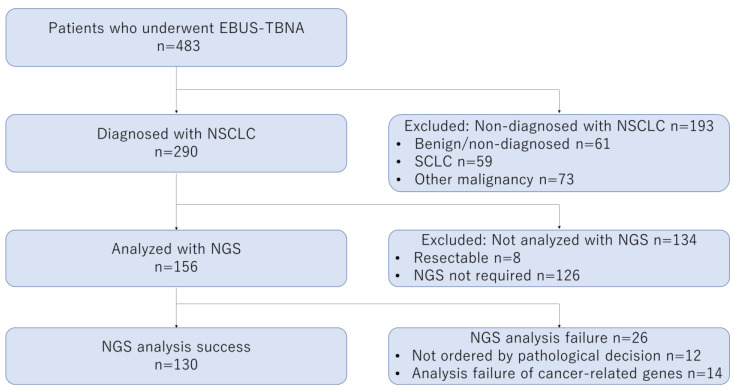
Flow diagram for next-generation sequencing (Oncomine^TM^ Dx Target Test). EBUS-TBNA, endobronchial ultrasound-guided transbronchial needle aspiration; NSCLC, non-small cell lung cancer; SCLC, small cell lung cancer; NGS, next-generation sequencing.

**Figure 3 cancers-13-05879-f003:**
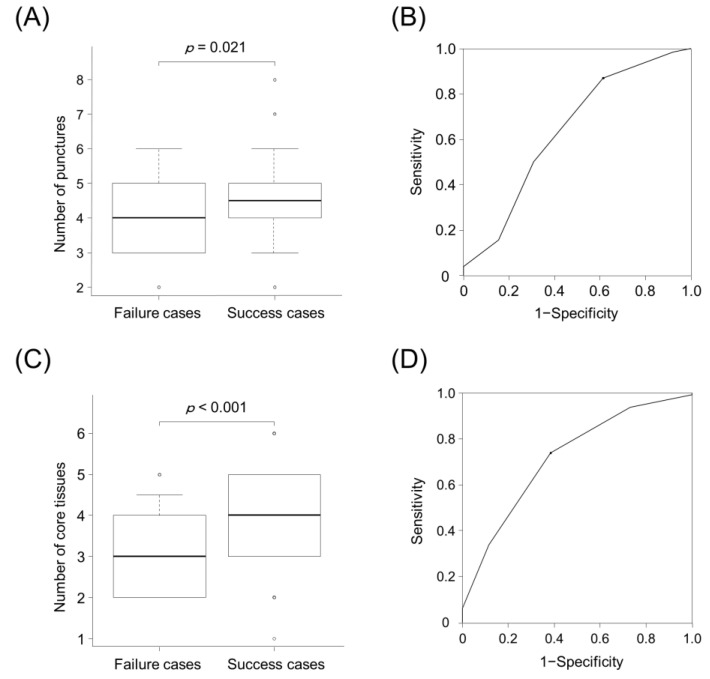
Number of punctures and core tissues in failure and success cases of NGS (Oncomine^TM^ Dx Target Test) analysis using specimens obtained through EBUS-TBNA. (**A**) Success cases showed a higher number of punctures than failure cases (5 vs. 4, *p* = 0.021, Mann–Whitney *U* test). (**B**) ROC curve analysis yielded a cut-off value of 4 for the number of punctures. Using this cut-off value, the sensitivity and specificity of the number of punctures for predicting successful NGS were 38.5% and 86.9%, respectively. The area under the curve was 0.637 (95% confidence interval, 0.51–0.765). (**C**) Success cases showed a higher number of core tissues than failure cases (4 vs. 3, *p* < 0.001, Mann–Whitney *U* test). (**D**) ROC curve analysis yielded a cut-off value of four for the number of core tissues. With this cut-off value, the sensitivity and specificity of the number of core tissues for predicting successful NGS were 61.5% and 73.8%, respectively. The area under the curve was 0.718 (95% confidence interval, 0.613–0.823). In (**A**) and (**C**), the bottom and top of the boxes indicate the 25th and 75th percentiles, respectively. The middle line of the boxes indicates the median. The whiskers indicate the 10th and 90th percentiles, and cases lying beyond these range are marked as circle dots.

**Figure 4 cancers-13-05879-f004:**
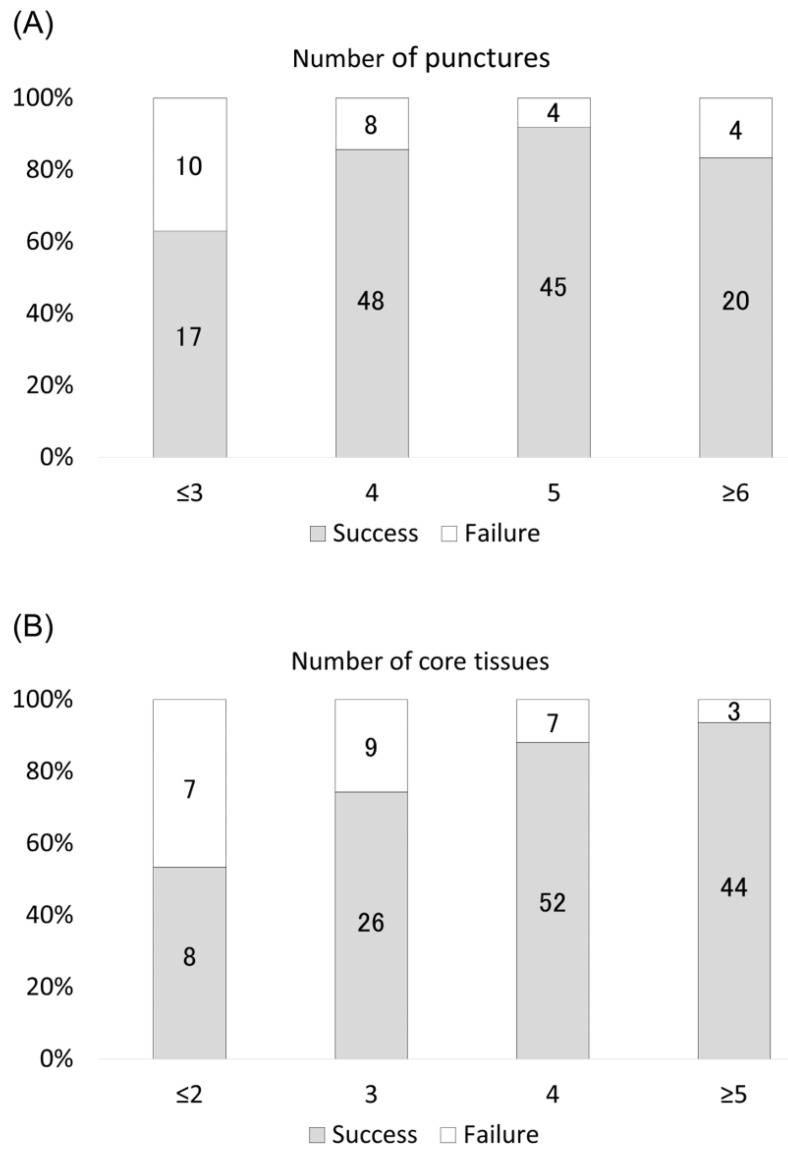
Success rate of NGS (Oncomine^TM^ Dx Target Test) with specimens obtained through EBUS-TBNA according to the (**A**) number of punctures and (**B**) core tissues. Higher numbers of punctures and core tissues were significantly associated with successful NGS (*p* = 0.028 and *p* < 0.001, Cochran–Armitage test). The number in the bar graph shows the number of cases.

**Table 1 cancers-13-05879-t001:** Cancer-related genes that can be analyzed with next-generation sequencing using the Oncomine^TM^ Dx Target Test.

Sample Type	Gene Name						
DNA	*AKT1*	*ALK*	*AR*	*BRAF*	*CDK4*	*CTNNB1*	*DDR2*
	*EGFR*	*ERBB2*	*ERBB3*	*ERBB4*	*ESR1*	*FGFR2*	*FGFR3*
	*GNA11*	*GNAQ*	*HRAS*	*IDH1*	*IDH2*	*JAK1*	*JAK2*
	*JAK3*	*KIT*	*KRAS*	*MAP2K1*	*MAP2K2*	*MET*	*MTOR*
	*NRAS*	*PDGFRA*	*PIK3CA*	*RAF1*	*RET*	*ROS1*	*SMO*
RNA	*ABL1*	*ALK*	*AXL*	*BRAF*	*ERBB2*	*ERG*	*ETV1*
	*ETV4*	*ETV5*	*FGFR1*	*FGFR2*	*FGFR3*	*MET*	*NTRK1*
	*NTRK2*	*NTRK3*	*PDGFRA*	*PPARG*	*RAF1*	*RET*	*ROS1*

**Table 2 cancers-13-05879-t002:** Baseline characteristics of patients who underwent next-generation sequencing using specimens obtained through endobronchial ultrasound-guided transbronchial needle aspiration.

Characteristics	All Cases	Success Cases	Failure Cases	*p* Value ^†^
	(*n* = 156)	(*n* = 130)	(*n* = 26)	
Age, years	70 (31–91)	70 (31–87)	70 (51–91)	0.54
Sex, male	103 (66.0)	84 (64.6)	19 (73.1)	0.50
Smoking history				0.47
Current or past	116 (74.4)	95 (73.1)	21 (80.8)	
Never	40 (25.6)	35 (26.9)	5 (19.2)	
Tumor clinical stage				0.81
II	3 (1.9)	3 (2.3)	0 (0)	
III	64 (41.0)	53 (40.8)	11 (42.3)	
IV	77 (49.4)	65 (50.0)	12 (46.2)	
Recurrence	12 (7.7)	9 (6.9)	3 (11.5)	-
Size of lymph nodes				0.45
Short axis on CT, mm	14.5 (5.2–66.8)	14.6 (5.2–66.8)	14.1 (7.5–43.7)	
Necrotic finding on CT				0.52
Positive	70 (44.9)	60 (46.2)	10 (38.5)	
Negative	86 (55.1)	70 (53.8)	16 (61.5)	
Location of lymph nodes				0.62
Upper paratracheal (2R, 2L, 3p)	11 (7.1)	9 (6.9)	2 (7.7)	
Lower paratracheal (4R, 4L)	60 (38.5)	51 (39.2)	9 (34.6)	
Subcarinal (7)	43 (27.6)	34 (26.2)	9 (34.6)	
Hilar (10R, 10L)	2 (1.3)	1 (0.8)	1 (3.8)	
Interlobar and lobar (11, 12)	19 (12.3)	16 (12.3)	3 (11.5)	
Others (lung, pleura)	21 (13.5)	19 (14.6)	2 (7.7)	
Pathological diagnosis				0.48
Adenocarcinoma	101 (64.7)	86 (66.2)	15 (57.7)	
Squamous cell carcinoma	21 (13.5)	18 (13.8)	3 (11.5)	
Other subtypes or not otherwise specified	34 (21.8)	26 (20.0)	8 (30.8)	
SUVmax on PET-CT				0.15
≥2.5	127 (81.4)	109 (83.8)	18 (69.2)	
<2.5	17 (10.9)	13 (10.0)	4 (15.4)	
Not evaluated	12 (7.7)	8 (6.2)	4 (15.4)	

Data are presented as number (%) or median (range). ^†^ Calculated using Fisher’s exact test or Mann–Whitney *U* test. CT, computed tomography; SUVmax, maximum standardized uptake value; PET, positron emission tomography.

**Table 3 cancers-13-05879-t003:** Logistic regression analysis of clinical factors affecting the success rate of next-generation sequencing using specimens obtained through EBUS-TBNA.

Variables	Univariable Analysis		Multivariable Analysis	
	Success Cases/All Cases (%)	*p* Value ^†^	Odds Ratio (95 %CI)	*p* Value ^‡^
Needle size		0.79	-	-
25-gauge	23/28 (82.1)			
22-gauge	107/128 (83.6)			
Systemic staging		1.0	-	-
Performed	41/49 (83.7)			
Not performed	89/107 (83.2)			
Necrotic finding on CT		0.52	-	-
Positive	55/64 (85.9)			
Negative	75/92 (81.5)			
B-mode categories				
Short axis, mm		0.62	-	-
≤10	32/37 (86.5)			
>10	98/119 (82.4)			
Shape		0.82	-	-
Round	42/51 (82.4)			
Oval	88/105 (83.8)			
Margin		0.63	-	-
Distinct	97/115 (84.3)			
Indistinct	33/41 (80.5)			
Echogenicity		1.0	-	-
Heterogeneous	128/154 (83.1)			
Homogeneous	2/2 (100)			
Central hilar structure		0.21	-	-
Presence	11/11 (100)			
Absence	119/145 (82.1)			
Coagulation necrosis sign		1.0	-	-
Presence	37/44 (84.1)			
Absence	93/112 (83.0)			
Color/power Doppler images		1.0	-	-
Vascular pattern II–III	76/91 (83.5)			
Vascular pattern 0–I	54/65 (83.1)			
Number of punctures		0.004	1.67 (0.49–5.62)	0.41
<4	17/27 (63.0)			
≥4	113/129 (87.6)			
Number of core tissues		<0.001	3.39 (1.09–10.6)	0.035
<4	34/50 (68.0)			
≥4	96/106 (92.5)			

Data are presented as number (%). ^†^ Calculated using Fisher’s exact test. **^‡^** Calculated using logistic regression analysis. CT, computed tomography.

## Data Availability

Data are available from the corresponding author upon reasonable request.

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
