# Peer review of "The Impact of Core Tissues on Successful Next-Generation Sequencing Analysis of Specimens Obtained through Endobronchial Ultrasound-Guided Transbronchial Needle Aspiration"

_cancers, 2021, doi:10.3390/cancers13235879_

Round 1
Reviewer 1 Report
The i/v sedation was moderate to deep. Was the EBUS performed (or considered to do) via rigid bronchoscope, which increases the sensitivity of the procedure. It should mentioned in the text.
The topic is important in the field. However it just proves everyday practise- we should obtain as much material (sample tissue) as possible for the analysis. It is written in good scientific language and well based on sources. Methodology is chosen by authors and is correct.
Author Response
Reviewer 1 Comments:
The i/v sedation was moderate to deep. Was the EBUS performed (or considered to do) via rigid bronchoscope, which increases the sensitivity of the procedure. It should mentioned in the text.
The topic is important in the field. However it just proves everyday practise- we should obtain as much material (sample tissue) as possible for the analysis. It is written in good scientific language and well based on sources. Methodology is chosen by authors and is correct.
Response:
We deeply appreciate your review, and your suggestions have helped us improve the quality of our manuscript. We specified that the EBUS procedures were performed with a flexible bronchoscope, not a rigid bronchoscope. Regarding your comment, we have revised our manuscript as follows and hope that our response sufficiently addresses your concern:
Revised sentence (changes are underlined):
Page 3, line 106
(Before revision)
“All patients underwent bronchoscopy under moderate-to-deep sedation with intravenous anesthesia.”
(After revision)
“All patients underwent flexible bronchoscopy under moderate-to-deep sedation with intravenous anesthesia.”

Reviewer 2 Report
The authors described here the impact of number of core tissues of EBUS TBNA on successful NGS analysis. They also showed that CT and EBUS findings were not associated with successful NGS.
This paper is not original and do not add any novelty in the field: more you have tissue (and so DNA) and more NGS will be successful...
It is clear cut that the important biological markers is not the number of core tissues but the DNA quantity which is clearly missing here.
The cut off of 4 core tissues is a good findings, but it is clear that it will depend on the bronchoscopist... Moreover the authors did not try to explain why some samples with 4 or more core tissues failed: DNA quantity? DNA quality? ...Likely, why some samples with 2 or 3 cores tissues succeeded and others failed?
Why the authors only looked at 4 cancer related genes using this panel of 46 genes? What about ERBB2 mutations? splice site mutation of exon 14 of MET? This Oncomine Dx Target Test multi CDx system kit look for fusion transcript on DNA or RNA? Important details are missing in part 2.3.
Author Response
Reviewer 2 Comments:
The authors described here the impact of number of core tissues of EBUS TBNA on successful NGS analysis. They also showed that CT and EBUS findings were not associated with successful NGS. This paper is not original and do not add any novelty in the field: more you have tissue (and so DNA) and more NGS will be successful...
It is clear cut that the important biological markers is not the number of core tissues but the DNA quantity which is clearly missing here.
Response:
We are deeply appreciative of your review and for the appropriate suggestions for improving the quality of our manuscript. However, to the best of our knowledge, no studies have examined whether imaging findings (CT and EBUS) and the number of core tissues are associated with success of NGS using EBUS-TBNA specimens. As you pointed out, the most important contributor to successful NGS is definitely the amount of good quality DNA and RNA that can be isolated from specimens collected through EBUS-TBNA. However, it is not possible to evaluate the content of the specimens during the actual EBUS-TBNA procedure. In addition, even if the amount of DNA and RNA in the tissue is sufficient, NGS may not always be successful due to quality-related issues. Therefore, in this study, we mainly focused on investigating whether the success of NGS can be predicted by the content of the tissue collected through the EBUS-TBNA procedure and the CT or EBUS imaging findings. As stated in the limitations paragraph of the Discussion section, we were unable to quantify the amount of DNA and RNA in this study. However, we still believe that our manuscript describes novel and clinically important findings, and we hope that our responses sufficiently address your concerns. Regarding your comments, we have provided point-by-point responses and descriptions of the revisions made to our manuscript as follows:
Point 1:
The cut off of 4 core tissues is a good findings, but it is clear that it will depend on the bronchoscopist... Moreover the authors did not try to explain why some samples with 4 or more core tissues failed: DNA quantity? DNA quality? ...Likely, why some samples with 2 or 3 cores tissues succeeded and others failed?
Response:
Thank you for your relevant comments. As you pointed out, there may be differences in the quantity and quality of the specimens collected by different bronchoscopists. In our institution, EBUS-TBNA is always performed under the supervision of experienced bronchoscopists in order to ensure the accuracy of the examination. Therefore, we have added a sentence to the Methods section of the manuscript.
Added sentence (revisions are underlined):
Page 3, lines 117–118
“All procedures were performed under the supervision of two bronchoscopists with more than 10 years of experience.”
In addition, we were unable to assess the quantity and quality of the DNA and RNA in the tissues in this study, and we could not determine why NGS failed for cases in which EBUS-TBNA specimens contained more than four core tissues or why NGS was successful for EBUS-TBNA specimens containing 2–3 core tissues. As we noted in the Discussion section (Page 10, lines 278–280), it has been previously reported that as many as 50–60% of EBUS-TBNA specimens per puncture are insufficient for diagnosis. We believe that this is an inherent limitation of the EBUS-TBNA technique for tissue sampling. Even when we obtained core tissues through EBUS-TBNA, we often found contamination of only blood, cartilage, or bronchial walls in the pathological specimens. Therefore, we believe that obtaining a sufficient number of core tissues through EBUS-TBNA is a relevant predictor of the success of NGS.
Point 2:
Why the authors only looked at 4 cancer related genes using this panel of 46 genes? What about ERBB2 mutations? splice site mutation of exon 14 of MET? This Oncomine Dx Target Test multi CDx system kit look for fusion transcript on DNA or RNA? Important details are missing in part 2.3.
Response:
We apologize for the lack of explanation about the ODxTT multi CDx system. This system was used to analyze 46 cancer-related genes in the patients; however, in this study, we only examined the results of the four cancer-related genes (EGFR, EML4/ALK, ROS1, and BRAF) that are approved for evaluation using the ODxTT multi CDx system for companion diagnostic testing in Japan, as described in the Introduction section.
The ODxTT multi CDx system is capable of analyzing mutations at hotspots in 46 cancer-related genes using DNA derived from tumor specimens, as well as fusions in 21 cancer-related genes using RNA derived from tumor specimens with amplicon sequencing. As you pointed out, our manuscript did not provide a sufficient description of the ODxTT multi CDx system and the 46 cancer-related genes that can be analyzed, so we have added or modified sentences in the Methods section and listed the 46 cancer-related genes that can be analyzed using this system in Table 1. The other tables have been renumbered accordingly.
Additions (changes are underlined):
Page 4, lines 139–141 and Table 1 (lines 142–144)
“The ODxTT multi CDx system can analyze mutations at hotspots in 46 cancer-related genes using DNA derived from tumor specimens, as well as fusions in 21 cancer-related genes using RNA derived from tumor specimens with amplicon sequencing (Table 1).”
Table 1. Cancer-related genes that can be analyzed with next-generation sequencing using the OncomineTM Dx Target Test.
|
DNA |
|
|
|
|
|
|
|
|
|
AKT1 |
ALK |
AR |
BRAF |
CDK4 |
CTNNB1 |
DDR2 |
|
|
EGFR |
ERBB2 |
ERBB3 |
ERBB4 |
ESR1 |
FGFR2 |
FGFR3 |
|
|
GNA11 |
GNAQ |
HRAS |
IDH1 |
IDH2 |
JAK1 |
JAK2 |
|
|
JAK3 |
KIT |
KRAS |
MAP2K1 |
MAP2K2 |
MET |
MTOR |
|
|
NRAS |
PDGFRA |
PIK3CA |
RAF1 |
RET |
ROS1 |
SMO |
|
RNA |
|
|
|
|
|
|
|
|
|
ABL1 |
ALK |
AXL |
BRAF |
ERBB2 |
ERG |
ETV1 |
|
|
ETV4 |
ETV5 |
FGFR1 |
FGFR2 |
FGFR3 |
MET |
NTRK1 |
|
|
NTRK2 |
NTRK3 |
PDGFRA |
PPARG |
RAF1 |
RET |
ROS1 |
Revised sentences (changes are underlined):
Page 4, lines 147–149
(Before revision)
“Success” was defined as cases where all four cancer-related genes (EGFR, EML4/ALK, ROS1, and BRAF) could be analyzed through NGS.
(After revision)
“The ODxTT multi CDx system is only approved for companion diagnostic testing for four cancer-related genes (EGFR, EML4/ALK, ROS1, and BRAF) in Japan. Therefore, in this study, “success” was defined as cases where all four cancer-related genes (EGFR, EML4/ALK, ROS1, and BRAF) could be analyzed through NGS.”

Reviewer 3 Report
Dear Authors,
I read with great interest the manuscript you submitted to Cancers.
It is an extremely interest study on a field of actuality, in particular about the personalized medicine era.
The manuscript is well designed and the amount of cases in considerable, looking at the study period.
I have some minor comments:
- I found the Introduction section too long and leading the reader to confusion. I suggest the authors to shorten it and focus directly on the object of the study
- It is not clear why and when the different needles are used (22 or 25 Gauges)
- Table 2: why multivariable analysis was performed only for Nr of puntures and Nr. of core tissues?
In conclusion, I found the manuscript of high value and with potential high impact on daily clinical practice for surgeons, pathologists, endoscopists and oncologists.
Best regards
Author Response
Reviewer 3 Comments:
Dear Authors,
I read with great interest the manuscript you submitted to Cancers.
It is an extremely interest study on a field of actuality, in particular about the personalized medicine era. The manuscript is well designed and the amount of cases in considerable, looking at the study period.
I have some minor comments:
- I found the Introduction section too long and leading the reader to confusion. I suggest the authors to shorten it and focus directly on the object of the study
- It is not clear why and when the different needles are used (22 or 25 Gauges)
- Table 2: why multivariable analysis was performed only for Nr of puntures and Nr. of core tissues?
In conclusion, I found the manuscript of high value and with potential high impact on daily clinical practice for surgeons, pathologists, endoscopists and oncologists.
Best regards
Response:
We deeply appreciate your review and suggestions, as they have helped us produce a better manuscript. Regarding your minor comments, we have provided point-by-point responses and described the revisions we have made to our manuscript below. We hope that our changes have sufficiently addressed your concerns.
Point 1:
I found the Introduction section too long and leading the reader to confusion. I suggest the authors to shorten it and focus directly on the object of the study
Response:
We apologize for the length of the Introduction section, and we agree that providing excessive or irrelevant information can be confusing to the readers. We considered shortening this section; ultimately, however, we decided that it would be difficult to remove sentences about the target genes for lung cancer therapy, NGS, EBUS-TBNA, and imaging findings (EBUS and CT), as all of these topics are important for establishing the aim of this study. We hope you will accept our decision not to remove these sentences.
Point 2:
It is not clear why and when the different needles are used (22 or 25 Gauges)
Response:
Thank you for pointing out this missing information related to the lumen diameter of the needles used. The decision to use a 25-G needle was at the operators’ discretion in cases in which systemic staging was required or when the approach was deemed to be too difficult with a 22-G needle. We have added the above information to the Methods section as follows:
Added sentence (changes are underlined):
Page 3, lines 112–114
Although 22-G needles were mainly used for EBUS-TBNA, 25-G needles were selected for cases that required systemic staging or for those in which the operator believed puncturing the tissue would be difficult using a 22-G needle.
Point 3:
Table 2: why multivariable analysis was performed only for Nr of puntures and Nr. of core tissues?
Response:
We apologize for failing to adequately describe the reasoning for this in the Statistical Analysis subsection of the Methods. No previous studies have investigated whether the content of tissues collected through EBUS-TBNA procedures or imaging findings (EBUS and CT) contributed to the success of NGS with EBUS-TBNA specimens. Therefore, in this study, multivariable analysis was performed on the clinical factors that were shown to influence the success of NGS in the univariable analysis. As you pointed out, the description was inadequate, so we have revised a sentence in that subsection of the Methods as follows:
Revised sentence (changes are underlined):
Page 5, lines 186–188
(Before revision)
“Further, we performed a multivariable logistic regression analysis of clinical factors influencing successful NGS.”
(After revision)
“Further, we performed a multivariable logistic regression analysis of the clinical factors that were shown to influence the success of NGS in the univariable analysis.”

Reviewer 4 Report
Recently I was invited to review a paper entitled “The impact of core tissues on successful next-generation sequencing analysis of specimens obtained through endobronchial ultrasound-guided transbronchial needle aspiration”. This manuscript discusses an important issue of advanced molecular testing of small tissue samples obtained by the means of EBUS. The topic chosen by the authors is important and up to date. I believe there is a significant citing potential of this paper.
I understand the need to confirm the main finding of the study – the requirement to perform at least 4 punctures or core biopsies to obtain an optimal material for NGS. Authors use the multivariable analysis and ROC curves analysis. On one hand, the result of the multivariable analysis is clear. On the other hand, the sensitivities disclosed by ROC curves are as low as 38% and 61% with AUC 0.637 and 0.718. This limited sensitivity disclosed in this analysis makes the results of the paper not that obvious. Please consider two approaches: discussing the poor results of ROC curves analysis in study limitations or resignation from presenting this aspect of analysis.
Anyway, I would like to congratulate the authors on this interesting and clearly presented paper.
Author Response
Reviewer 4 Comments:
Recently I was invited to review a paper entitled “The impact of core tissues on successful next-generation sequencing analysis of specimens obtained through endobronchial ultrasound-guided transbronchial needle aspiration”. This manuscript discusses an important issue of advanced molecular testing of small tissue samples obtained by the means of EBUS. The topic chosen by the authors is important and up to date. I believe there is a significant citing potential of this paper.
I understand the need to confirm the main finding of the study – the requirement to perform at least 4 punctures or core biopsies to obtain an optimal material for NGS. Authors use the multivariable analysis and ROC curves analysis. On one hand, the result of the multivariable analysis is clear. On the other hand, the sensitivities disclosed by ROC curves are as low as 38% and 61% with AUC 0.637 and 0.718. This limited sensitivity disclosed in this analysis makes the results of the paper not that obvious. Please consider two approaches: discussing the poor results of ROC curves analysis in study limitations or resignation from presenting this aspect of analysis.
Anyway, I would like to congratulate the authors on this interesting and clearly presented paper.
Response:
We deeply appreciate your review and the appropriate suggestions for clarifying the meaning of our results. As you pointed out, the results of the ROC curve analysis of the number of punctures and core tissues were insufficient. However, this study did not identify any clinical factors that contributed to successful NGS other than the number of core tissues. Therefore, we believe that the number of core tissues obtained through EBUS-TBNA should be focused on for ensuring successful NGS using EBUS-TBNA specimens. Regarding your suggestions, we added several sentences pertaining to the poor results of the ROC curve analysis in the limitations of the Discussion section as follows. We hope that our response will assuage your concerns about the sensitivity.
Added sentence (changes are underlined):
Page 11, lines 329–333
“Fifth, the results of the ROC curve analysis of the number of punctures and core tissues were insufficient. However, no clinical factors contributing to the success of NGS other than the number of core tissues were evident in this study. Therefore, the number of core tissues obtained through EBUS-TBNA should be focused on for ensuring successful NGS.”

Round 2
Reviewer 2 Report
Thanks to the authors for their answers.
However, these answers are not pertinent and many informations are lacking.
We would like to thank the reviewer for their constructive critique to improve the manuscript. We have made every effort to address the issues raised and to respond to all comments. We would like to apologize to the reviewer that our previous revisions did not address his/her concerns. Please find next a detailed, point-by-point response to the reviewer's comments. We hope that our revisions would meet their expectations.
How do the DNA and RNA are extracted? After extraction, every laboratory measure the quantity of DNA and RNA to know the volume they will need to pursue the technique.
Otherwise, an important part about DNA /RNA extraction is missing here...
Response:
We would like to thank the reviewer for the comment. We would like to apologize for not providing this important information. Please note that we have added more information concerning DNA and RNA extraction procedures as follows:
Added sentence (changes are underlined):
Page 4, lines 147–158
(After revision)
“DNA and RNA extractions were performed using the Ion Torrent Dx FFPE Sample Preparation Kit, which was included as part of the ODxTT multi CDx system for the applied biosystems VRTi Dx (Thermo Fisher Scientific). The deparaffinized samples from FFPE specimens were lysed using protease and filter cartridges were used to separate DNA and RNA. DNA and RNA inputs were 10 ng each and were measured using the NanoDrop spectrophotometer (Thermo Fisher Scientific) and the Qubit® 2.0 fluorometer (Thermo Fisher Scientific). Library preparation was performed using the ODxTT multi CDx system, controls, and Ion PGM Dx Library Kit. Template preparation was performed using Ion One Touch Duo Dx and Ion One Touch Dx Template Supplies, and Ion PGM Dx Sequencing Kit was used for sequencing platform. Mutations among cancer-related genes were analyzed by the Torrent Suite Dx software (Thermo Fisher Scientific).”
Did the authors noticed that samples with 4 cores tissues failed because of blodd, necrosis, cartilage.... ? Did the authors noticed than samples with 2 or 3 cores tissues succeeded because of important tumor cellularity?
Response:
We would like to thank the reviewer for the questions. In this study, we did not examine in detail the tumor content ratio and the degree of blood contamination in the EBUS-TBNA specimens of all cases. However, we re-examined the histological results for 10 cases where NGS analysis failed, despite the availability of four or more core tissues. In the six cases where histological specimens showed very little tumor tissue and were not submitted for the NGS analysis, almost all of the core tissue was blood component. In the four cases that were submitted to NGS analysis (tumor cells per slide >5000 cells), there was a lot of blood contamination, and the tumor content ratio was <20%. Although it may be difficult to conclude as these results were not reviewed histologically in all cases, our findings highlighted blood contamination in the EBUS-TBNA specimens, thus resulting in the failure of the NGS analysis.
As the reviewer pointed out, the reason why NGS analysis was successful even in cases where only two or three core tissues were obtained might be due to the fact that we obtained enough tumor cells via EBUS-TBNA. However, in this study, the number of tumor cells was not counted in detail and, therefore, we cannot derive clear conclusions.
Round 3
Reviewer 2 Report
Thank you for letting me consider the new responses of the authors.
The authors answers all my questions/comments.
Yes i approve the paper for publication.